# VISION TRANSFORMER FOR MULTIVARIATE TIME-SERIES CLASSIFICATION (VITMTSC)

## ABSTRACT

Multivariate Time-Series Classification (MTSC) is an important issue in many disciplines because of the proliferation of disparate data sources and sensors (economics, retail, health, etc.). Nonetheless, it remains difficult due to the high-dimensionality and richness of data that is regularly updated. We present a Vision Transformer for Multivariate Time-Series Classification (VitMTSC) model that learns latent features from raw time-series data for classification tasks and is applicable to large-scale time-series data with millions of data samples of variable lengths. According to our knowledge, this is the first implementation of the Vision Transformer (ViT) for MTSC. We demonstrate that our approach works on datasets ranging from a few thousand to millions of samples and achieves close to the state-of-the-art (SOTA) results on open datasets. Using click-stream data from a major retail website, we demonstrate that our model can scale to millions of samples and vastly outperform previous neural net-based MTSC models in real-world applications. Our source code is publicly accessible at `https://github.com/mtsc-research/vitmtsc` to facilitate further research.

## 1 INTRODUCTION

Deep neural networks (DNNs) have shown significant effectiveness with both text (Lin et al., 2021) and images (Li et al., 2021). The availability of standard DNN architectures that effectively encode raw data into meaningful representations, leading in good performance on new datasets and associated tasks with little effort, is a crucial factor enabling advancement. In image understanding, for instance, variations of residual convolutional networks, such as ResNet (He et al., 2015), exhibit relatively outstanding performance on new image datasets or somewhat different visual recognition problems, such as segmentation. Despite being the most prevalent sort of data in the actual world, tabular data has only recently been the focus of deep learning research, such as TabNet (Arik & Pfister, 2019) and TabTransformer (Huang et al., 2020). Time-Series data is a special case of tabular data. An intrinsic characteristic of time-series is that observations within time-series are statistically dependent on assumed generative processes (Löning et al., 2019). For example, the likelihood of a user streaming a particular movie $X$, is dependent on whether they have streamed movie $X$ before as well as latent factors like, the time duration since the last streaming of movie $X$, number of movies from the same genre/director/actor as $X$ that the user has streamed, cost of streaming, promotions on movie $X$, etc. Due to this reliance, time-series data does not readily fit inside the traditional machine learning paradigm for tabular data, which implicitly implies that observations are independent and identically distributed (i.i.d.).

We consider time-series, input to the model, to be composed of (a) time-points at which they are observed, and (b) observations (or time-series datapoints or TS datapoints) at those time-points. We denote, a time-series object that contains $N$ samples as $\mathbf{X} = [\mathbf{X}_{t_1}, \mathbf{X}_{t_2}, ..., \mathbf{X}_{t_N}]$, where elements of the sequence are observation(s) at time-points $t_1$, $t_2$, ..., $t_N$ respectively. The time-series data may be (a) Univariate time-series (UTS), in which a single variable is observed over time, or (b) Multivariate time-series (MTS), in which two or more variables are recorded over time. We denote an individual multivariate time-series datapoint as $\mathbf{X}_{t_K} = [\mathbf{X}_{t_K}^1, \mathbf{X}_{t_K}^2, ..., \mathbf{X}_{t_K}^M]^T$ for 1, 2, ..., $M$ distinct variables/features observed at each time-point. Please note in large commercial datasets, an individual $\mathbf{X}_{t_K}^n$ categorical variable can have millions of distinct values (refer to Table 2).

In this paper, we will focus on Multivariate Time-Series Classification (MTSC) (Ruiz et al., 2021), which is an area of machine learning interested in assigning labels to MTS data. In non-DNN based MTSC methods, the data is first converted to i.i.d. via feature transformation, and then traditional

classifiers are applied (Ruiz et al., 2021). Only a few available approaches for MTSC consider DNNs for this task (Fawaz et al., 2018) (refer to Section 2). One advantage of using DNN for MTSC is predicted efficiency benefits, particularly for large datasets (Hestness et al., 2017). In addition, DNNs offer gradient descent (GD) based end-to-end learning for time-series data, which has numerous advantages: (a) quickly encoding multiple data kinds, including images, tabular, and time-series data; (b) eliminating the requirement for feature engineering, which is a fundamental part of non-DNN MTSC approaches; (c) learning from streaming data; and perhaps most crucially (d) end-to-end models enable representation learning, which enables many valuable applications including including data-efficient domain adaptation (Goodfellow et al., 2016), generative modeling (Radford et al., 2015), semi-supervised learning (Dai et al., 2017), and unsupervised learning (Karhunen et al., 2015).

We present Vision Transformer for Multivariate Time-Series Classification (**VitMTSC**), a ViT-based pure transformer approach to MTSC, to model long-range contextual relationships in MTS data. The VitMTSC model accepts raw time-series data as input and is trained with GD based optimization to facilitate flexible end-to-end learning. We evaluate VitMTSC using both publicly available and proprietary datasets. The majority of algorithms in the MTSC literature have been evaluated on open-source UEA datasets (Ruiz et al., 2021). We evaluate VitMTSC on five UEA datasets (Table 1) and demonstrate that VitMTSC achieves comparable performance to the current SOTA methods (Table 3). Using click-stream data (Table 2) from a popular e-commerce site, we show that VitMTSC can meet the needs of modern datasets and performs much better than current DNN methods (Table 4) on both small and large real-world datasets (refer to Section 4 for experiment details).

## 2 RELATED WORK

Since AlexNet (Krizhevsky et al., 2012), deep convolutional neural networks (CNNs) have advanced the state-of-the-art across many standard datasets for vision problems. At the same time, the most prominent architecture of choice in sequence-to-sequence modeling is the Transformer (Vaswani et al., 2017), which does not use convolutions, but is based on multi-headed self-attention (MSA). The MSA operation is particularly effective at modelling long-range dependencies and allows the model to attend over all elements in the input sequence. This is in stark contrast to convolutions where the corresponding "receptive field" is limited, and grows linearly with the depth of the network. The success of attention-based models in Natural Language Processing (NLP) has inspired approaches in computer vision to integrate transformers into CNNs (Wang et al., 2017; Carion et al., 2020), as well as some attempts to replace convolutions completely (Parmar et al., 2018; Bello et al., 2019; Ramachandran et al., 2019). With the Vision Transformer (ViT) (Dosovitskiy et al., 2020), however, a pure-transformer-based architecture has recently outperformed its convolutional counterparts in image classification. Since then, the ViT model has also been applied to other domains, for example, video classification (Arnab et al., 2021) and many more variants have been proposed [1].

In the last two decades, time-series data has proliferated a plethora of fields including economics (Wan & Si, 2017), health and medicine (Gharehbaghi et al., 2015), scene classification (Nwe et al., 2017), activity recognition (Chen et al., 2020), traffic analysis (Yang et al., 2014), click-stream analysis (Jo et al., 2018) and more. DNN is used in modeling time-series tasks, such as forecasting (Benidis et al., 2020; Lim & Zohren, 2021; Torres et al., 2021), classification (Wang et al., 2016; Fawaz et al., 2018; Zou et al., 2019; Fawaz et al., 2019), anomaly detection (Blázquez-García et al., 2020; Choi et al., 2021), and data augmentation (Wen et al., 2021). Similarly, Transformers (Wen et al., 2022) have been used in Time-Series forecasting (Li et al., 2019; Zhou et al., 2020; 2022), anomaly detection (Xu et al., 2021; Tuli et al., 2022), classification (Rußwurm et al., 2019; Zerveas et al., 2021; Yang et al., 2021).

Historically, the problem of MTSC was addressed by non-linear transforms of the time-series on which standard classification algorithms are used. Bostrom & Bagnall (2017) used a modification of the shapelet transform to quickly find $k$ shapelets of length $n$ to represent the time-series. They then use the $kxn$ feature vector in a standard classification model. Similarly, BagOfPatterns (Lin et al., 2012) use the frequency of each word (token/feature) in the series to represent the series as a histogram of words. Bag of Symbolic Fourier Approximation Symbols (BOSS) (Large et al., 2018) uses Symbolic Fourier Approximation on frequencies of words in the series. Word ExtrAction for time SEries cLassification (WEASEL) (Schäfer & Leser, 2017) extracts words with multiple sliding windows of different sizes and selects the most discriminating words according to the chi-squared

---

[1] https://github.com/lucidrains/vit-pytorch

test. RandOm Convolutional KErnel Transform (ROCKET) (Dempster et al., 2019) generates a large number (10k) random convolutional kernels and extracts two features from the convolutions: the maximum and the proportion of positive values. It's originally designed for univariate time-series classification (UTSC) and later adapted to MTSC. Though ROCKET shows great performance on smaller MTS benchmark datasets, using it for datasets with hundreds of millions of MTS data is impractical (refer to 4.3).

DNN approaches for MTSC have borrowed heavily from image processing models - mainly Convolutional Neural Nets (CNN). Wang et al. (2016); Lin et al. (2020) used attention based CNNs, while Fawaz et al. (2019) used an ensemble of CNNs and Zou et al. (2019); Fawaz et al. (2018) integrated residual networks with CNNs. Similarly, Transformers were used by Yang et al. (2021) for UTSC. Rußwurm et al. (2019); Zerveas et al. (2021) used transformer for the MTSC, and we have compared and contrasted their approaches with our proposed approach (refer to 4.6).

## 3 METHOD

### 3.1 VISION TRANSFORMER FOR MULTIVARIATE TIME-SERIES CLASSIFICATION (VITMTSC)

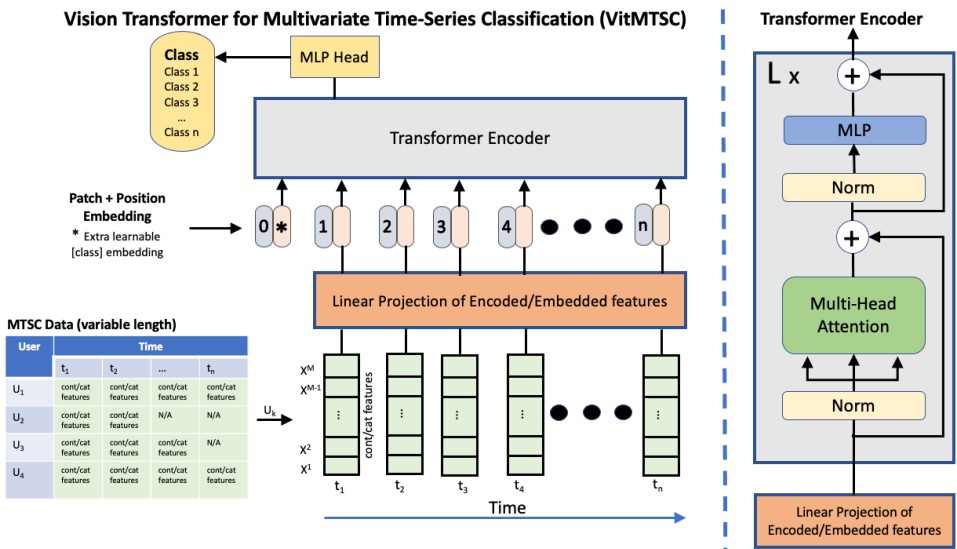

Figure 1: Vision Transformer for Multivariate Time-Series Classification (VitMTSC) - adapted from ViT (Dosovitskiy et al., 2020).

Figure 1 gives an overview of the VitMTSC architecture. We follow the original ViT design as closely as possible to take advantage of its scalable architecture, efficient implementations and future innovations (e.g., Bao et al. (2022)). Many variants of ViT has been proposed [2] but we will describe changes made to the original ViT model to handle MTSC.

### 3.1.1 TOKENIZATION: HOW TIME-SERIES DATA IS ENCODED FOR TRANSFORMER ENCODER?

The original Transformer architecture uses 1D sequence of token-ids as input, where a token-id is index of a word or sub-word in the vocabulary ($V$), and number of elements in this 1D sequence is the input sequence length. In the ViT, 2D images are first divided into multiple smaller 16x16 2D patches, and then each patch is flattened as 1D vector. This 1D vector is the input to the transformer, and total number of image patches serve as the effective input sequence length. Similar to the standard ViT, we start with MTS data, $\mathbf{X}$, and the number of time-steps in the MTS gives us input sequence length, $N$ for the Transformer. However, though input to a ViT is strictly numerical (pixel values), an MTS datapoint can contain a combination of continuous and categorical features (Table 2). There are multiple ways to encode continuous and categorical features. Each individual feature of an MTS datapoint is separately encoded to produce corresponding 1D vector. Then these encoded 1D vectors

---

[2]https://github.com/lucidrains/vit-pytorch

are aggregated (concatenated) and presented in 1D format as input to the VitMTSC transformer encoder.

**Feature Encoding.** There are different ways of encoding features. An individual continuous numerical feature of an MTS datapoint can be represented as a real-valued scalar, as a linear projection from a scalar, or as a Soft One-Hot Encoding (SOHE) (Li et al., 2018; de Souza Pereira Moreira et al., 2021). Categorical features in real-world datasets can take on millions of distinct values. Thus, encoding like one-hot encoding that use vectors proportional to the cardinality of the feature values, do not work for commercial datasets. Instead we use target encoding (Rodríguez et al., 2018). For each categorical feature, the mean of a continuous target column is calculated, and the group-specific mean of each row is used to create a new feature (column). To prevent overfitting, the mean is computed via k-fold cross-validation and smoothed with the overall mean of the target variable. This transformation results in a single feature for each categorical variable; not increasing the dimensionality of the dataset. In our dataset, we normalized continuous numerical scalar features and target encoded categorical features (NVIDIA, 2022). This helped keep model complexity low, in comparison to using corresponding embeddings (e.g., Patel & Bhattacharyya (2017) shows that embedding dimension lower than 19 are not performant and our corresponding commercial embedding size is at least 256).

**Padding and Truncation.** Since time-series data can have variable length, we post-pad the input array with 0s to generate sequences of a pre-defined length $N$. Depending on the application, longer sequences are truncated to either the first or last $N$ datapoints or use NLP style chunking (Briggs, 2021). We choose $N$ to be the $99^{th}$ percentile length in the dataset in our real-world application(Table 2).

### 3.1.2 TOKEN EMBEDDING

The original Transformer receives as input a 1D sequence of NLP tokens, map it to D dimensions with a trainable linear projection, and uses this constant latent vector size D through all of its layers. To handle 2D images, ViT, reshapes the original image in multiple smaller *patches*. They then flatten each patch and embed it as $D$-dimensional vector with a trainable linear projection. Similar to ViT, we embed the input sequence with a trainable linear projection ($\mathbf{E}$ in Eq. 1). The token embedding learns to represent each MTS datapoint as a vector in $\mathbb{R}^{d_E}$. Similar to BERT's [CLS] and ViT's [class] token, VitMTSC prepends a learnable embedding to the sequence of embedded patches (time-steps) ($\mathbf{z}_0^0 = \mathbf{x}_{class}$), whose state at the output of the Transformer encoder serves as the representation of the time-series. In addition, to patch embeddings we also use positional embeddings ($\mathbf{E_{pos}}$). We discuss positional embeddings in Section 3.1.3.

$$\mathbf{z}_0 = [\mathbf{x}_{class}; \mathbf{X}_{t_0}\mathbf{E}; \mathbf{X}_{t_1}\mathbf{E}; ...; \mathbf{X}_{t_N}\mathbf{E}] + \mathbf{E}_{pos} \qquad (1)$$

### 3.1.3 POSITIONAL EMBEDDING

Time-series data is ordered but Transformers are permutation invariant without positional information. The original Transformer (Vaswani et al., 2017) uses hand-crafted *absolute positional encoding*. Since the relative time between the data-points can be important for MTSC, we add *learnable 1D position embeddings* ($\mathbf{E_{pos}}$; see Algorithm 2 of Phuong & Hutter (2022)) to the time-step (patch) embeddings to retain this positional information before feeding input to Transformer encoder layer. The positional embedding is added to the token embedding to form a token's initial embedding for the transformer encoder (Eq. 1).

**Positional Encoding as Timestamp Features** Some datasets can have a set of (unordered) data points with the same timestamp. For example, for time-ordered set of products purchased by a customer, a single customer cart (same timestamp) may include multiple products. In such a case, absolute or learnable positional encodings will impose an arbitrary order on these datapoints (products). To avoid such an arbitrary order we do not use positional encoding on such datasets. Zhou et al. (2020) added both learnable and timestamp positional encoding to token embedding. Instead, we include the timestamp information (month, day, hour, etc of the cart purchase) of the datapoints as categorical features in the time-series. We found that the VitMTSC is able to use this information better than an arbitrary order imposed by the learnable positional encoding (refer Table 5).

### 3.1.4 ATTENTION

Attention is the main architectural component of transformers; see Algorithm 3 of Phuong & Hutter (2022). In practice, transformers use multi-headed self-attention (MSA) – multiple attention heads

(with separate learnable parameters) run in parallel – and combine their outputs; see Algorithm 5 of Phuong & Hutter (2022). VitMTSC also uses padding mask – it first converts 0s to 1s, and non-0 values to 0s, and then multiplies the padding mask with a large negative value. In VitMTSC's encoder MSA, the padding mask is added to the scaled score so that the original empty locations are now replaced with large negative values which will be eventually ignored by softmax activation function; see Algorithm 4 of Phuong & Hutter (2022), called with $\mathbf{Z} = \mathbf{X}$ and Mask.

### 3.1.5 LAYER NORMALIZATION

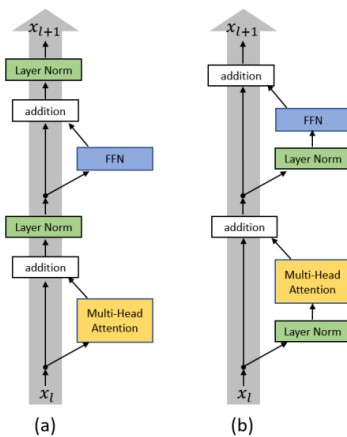

Figure 2: (a) Post-LN Transformer layer; (b) Pre-LN Transformer layer. - adapted from Xiong et al. (2020).

Layer Normalization (Ba et al., 2016) explicitly controls the mean and variance of individual neural network activations; see Algorithm 4 of Phuong & Hutter (2022). The original Transformer uses Layer Normalization **after** "MSA/FFN block + Residual connection" in the transformer encoder – called *POSTNORM*. Recently, GPT-2 (Radford et al., 2019), ViT (Dosovitskiy et al., 2020) and the ViTMTSC, use Layer Normalization just **before** "MSA/FFN block" in the transformer encoder – called *PRENORM* (Figure 2).

### 3.1.6 ENCODER-ONLY TRANSFORMER AND MODEL PREDICTION

Like ViT, the VitMTSC transformer encoder consists of alternating layers of multiheaded self-attention (MSA) + residual connection and multilayer perceptron (MLP) + residual connection. The MLP contains two layers with a Gaussian Error Linear Unit (GELU) (Hendrycks & Gimpel, 2016) non-linearity. Layernorm (LN) is applied just before MSA/MLP block and then residual connections are applied (Figure 2). VitMTSC's [class] token state at the output of the Transformer encoder ($\mathbf{z}_L^0$) serves as the time-series representation $\mathbf{y}$ (Eq. 2). The classification head is implemented by a multi-layer perceptron (MLP) with one hidden layer and is attached to $\mathbf{y}$ — the [class] token output of the transformer encoder; see detailed Algorithm 1, which has been adapted from Algorithm 8 of Phuong & Hutter (2022) [3].

$$\mathbf{y} = \text{LN}(\mathbf{z}_L^0) \tag{2}$$

## 4 EXPERIMENTS

We evaluate VitMTSC model along with current SOTA MTSC models on both publicly available and proprietary datasets.

---

[3]All current transformer models are some variation of the original Transformer. To set a new standard in DL publishing, Phuong & Hutter (2022) authors explicitly encourage the reader to copy and adapt them to their needs and cite the original as "adapted from [PH22]".

**Algorithm 1:** $P \leftarrow$ ETransformer($\boldsymbol{x}|\theta$) - adapted Algorithm 8 from Phuong & Hutter (2022)

**Input:** $\boldsymbol{x} \in V^*$, an MTS sequence. Let $V$ denote a set, such that $\mathbf{X}_{t_K} \in V$, often identified with $[N_V] := \{1, ..., N_V\}$. Let $\mathbf{X} \equiv \boldsymbol{x} \equiv x[1:\ell] \equiv x[1]x[2]...x[\ell] \in V^*$.

**Output:** $\boldsymbol{p} \in \Delta(C)$, a probability distribution over the classes in task.

**Hyperparameters:** $\ell_{max}, L, H, d_e, d_{mlp}, d_f \in \mathbb{N}$

**Parameters:** $\theta$ includes all of the following parameters:

$\boldsymbol{W_e} \in \mathbb{R}^{d_e \times N_V}, \boldsymbol{W_p} \in \mathbb{R}^{d_e \times \ell_{max}}$, the token embedding and positional embedding matrices.

For $l \in [L]$ :

$\quad \boldsymbol{W_l}$, multi-head attention parameters for the layer $l$,

$\quad \gamma_l^1, \boldsymbol{\beta}_l^1, \gamma_l^2, \boldsymbol{\beta}_l^2 \in \mathbb{R}^{d_e}$, two sets of layer-norm parameters,

$\quad \boldsymbol{W}_{mlp1}^l \in \mathbb{R}^{d_{mlp} \times d_e}, \boldsymbol{b}_{mlp1}^l \in \mathbb{R}^{d_{mlp}}, \boldsymbol{W}_{mlp2}^l \in \mathbb{R}^{d_{mlp} \times d_e}, \boldsymbol{b}_{mlp2}^l \in \mathbb{R}^{d_{mlp}}$, MLP parameters.

$\boldsymbol{W}_{mlp3}^{out1} \in \mathbb{R}^{d_e \times d_{mlp}}, \boldsymbol{b}_{mlp3}^{out1} \in \mathbb{R}^{d_{mlp}}, \boldsymbol{W}_{mlp4}^{out2} \in \mathbb{R}^{d_e \times d_{mlp}}, \boldsymbol{b}_{mlp4}^{out2} \in \mathbb{R}^{d_{mlp}}$, the final linear projection and layer-norm parameters.

1    $\ell \leftarrow$ length($\boldsymbol{x}$)

2    for $t \in [\ell] : \boldsymbol{e}_t \leftarrow \boldsymbol{W}_e[:, x[\text{t}]] + \boldsymbol{W}_p[:, \text{t}]$

3    $\boldsymbol{X} \leftarrow [\boldsymbol{e}_1, \boldsymbol{e}_2, ...\boldsymbol{e}_l]$

4    for $l = 1, 2, ..., L$ do

5      for $t \in [\ell] : \tilde{\boldsymbol{X}}[:, t] \leftarrow$ layer_norm($\boldsymbol{X}[:, t], \gamma_l^1, \boldsymbol{\beta}_l^1$)

6      $\boldsymbol{X} \leftarrow \boldsymbol{X} +$ MHAttention $(\tilde{\boldsymbol{X}} \mid \boldsymbol{W}_l, \text{Mask} \equiv 1)$

7      for $t \in [\ell] : \tilde{\boldsymbol{X}}[:, t] \leftarrow$ layer_norm($\boldsymbol{X}[:, t], \gamma_l^2, \boldsymbol{\beta}_l^2$)

8      $\boldsymbol{X} \leftarrow \boldsymbol{X} + \boldsymbol{W}_{mlp2}^l$ GELU($\boldsymbol{W}_{mlp1}^l \tilde{\boldsymbol{X}} + \boldsymbol{b}_{mlp1}^l \mathbf{1}^{\mathbf{T}}$) $+ \boldsymbol{b}_{mlp2}^l \mathbf{1}^{\mathbf{T}}$

9    **end**

10   $\boldsymbol{X} = \boldsymbol{X}[:, 1]$, discard everything and keep only [class] token.

11   for $t \in [\ell] : \boldsymbol{X}[:, t] \leftarrow$ layer_norm($\boldsymbol{X}[:, t], \boldsymbol{\gamma}, \boldsymbol{\beta}$)

12   $\boldsymbol{X} \leftarrow \boldsymbol{X} + \boldsymbol{W}_{mlp4}^{out2}$ GELU($\boldsymbol{W}_{mlp3}^{out1} \boldsymbol{X} + \boldsymbol{b}_{mlp3}^{out1} \mathbf{1}^{\mathbf{T}}$) $+ \boldsymbol{b}_{mlp4}^{out2} \mathbf{1}^{\mathbf{T}}$

13   **return** $\boldsymbol{p}$ = softmax($\boldsymbol{X}$)

Table 1: Open-source UEA datasets used for evaluation. It has only continuous (CONT) features and time-series datapoints are combined across both train and test datasets.

| DATASET | LENGTH | CONT | CAT | CLASSES | TRAIN | TEST | TS DATAPOINTS |
|---------|--------|------|-----|---------|-------|------|---------------|
| PenDigits | 8 | 2 | N/A | 10 | 7,494 | 3,498 | 87,936 |
| SpokenArabicDigits | 4-93 | 13 | N/A | 10 | 6,599 | 2,199 | 350,253 |
| CharacterTrajectories | 60-182 | 3 | N/A | 20 | 1,422 | 1,436 | 342,939 |
| FaceDetection | 62 | 4 | N/A | 6 | 5,890 | 3,524 | 583,668 |
| InsectWingbeat | 2-22 | 200 | N/A | 10 | 25,000 | 25,000 | 335,534 |

## 4.1 DATA

The majority of algorithms in the MTSC literature have been evaluated on open-source UEA datasets (Ruiz et al., 2021). The individual datasets (Table 1) in the UEA datasets are very small (the largest dataset has only 50k time-series data). Modern commercial datasets on the other hand contain hundreds of millions of time-series datapoints (refer Table 2). Also, UEA datasets only contain continuous variables, whereas real-world datasets can include a mix of continuous, discrete, and categorical variables. Though we evaluate VitMTSC on UEA datasets, the real power of the method is seen in real-world data.

### 4.1.1 OPEN-SOURCE UEA DATASETS

The UEA MTSC archive is a repository of benchmark datasets for time-series classification (Bagnall et al., 2018). It includes 30 multivariate datasets, four of which have variable lengths. We test our model on the five datasets (refer Table 1) available in the UEA repository, which include three datasets of variable length, a condition frequently observed in real-world data.

### 4.1.2 REAL-WORLD COMMERCIAL DATASETS

We used VitMTSC and other DNN MTSC models to analyze clickstream data on 9 different stores of a popular e-retailer. The aim of our study was to identify the set of customers that would be responsive

Table 2: Commercial datasets with both continuous (CONT) and categorical (CAT) features

| DATASET | LENGTH | CONT | CAT | CLASSES | TRAIN | VALID | TEST | TS DATAPOINTS |
|---------|--------|------|-----|---------|-------|-------|------|---------------|
| Small | 5-300 | 8 | 20 | 2 | 595K | 198K | 198K | 94M |
| Large | 10-300 | 8 | 20 | 2 | 4.2M | 0.9M | 0.9M | 922M |

to specific marketing campaigns. Using click-stream data as input, we build a binary classifier with the target variable being respond/not respond to a marketing campaign. We use response data from similar historical marketing campaigns for each retail store and train a separate model for each store. One of our smallest store has 992k time-series while the largest one has 6m. Each data point in the time-series consists of 20 features – 8 continuous and 12 categorical – that include products that the customer has clicked/purchased, product category, interaction type (click/purchase), time features (month, day, hour, etc), product attributes (price, discount, etc), customer attributes and more. Also, the space of values that a categorical variable can take can be quite large (e.g.: product ID can be one of hundreds of thousands to hundreds of millions of products depending on store).

### 4.2 RESULTS

### 4.3 BENCHMARK DATASETS

For MTSC, ROCKET (Dempster et al., 2019), and CMFMTS (Baldan & Benítez, 2020) are current SOTA Non-DNN methods and TST (Zerveas et al., 2021) is current SOTA DNN method. On the datasets of interest to us, they outperform all other models. Since UEA datasets specify both train and test datasets, we are able to use performance numbers from Baldan & Benítez (2020) and Zerveas et al. (2021) as CMFMTS and TST performances respectively. Unlike current practice in MTSC research(Section 6.6 of Fawaz et al. (2019), and Section 4 of Zerveas et al. (2021)), we further divided train data in 80-20% split and use this new validation dataset to tune both hyper-parameters and guide training to avoid overfitting on test dataset. We use provided test dataset only in the end to check model generalization capabilities, like we do in commercial applications. Table 3 gives the average accuracy of CMFMTS, TST, ROCKET and VitMTSC on these five open-source MTSC datasets.

ROCKET is one of the most prominent time-series classification algorithm. Unlike Zerveas et al. (2021), we were able to train ROCKET on all open-source datasets using sktime implementation[4]. However, on a p3dn.24xlarge (768 GB RAM and 96 CPUs) machine, ROCKET took 2 hours 40 minutes to complete training/inference on the biggest open-source dataset, InsectWingbeat (Table 1) that has only 50K time-series datapoints. This huge run-time renders it unusable in commercial datasets (Table 2) with millions of time-series datapoints and we won't discuss it further.

Table 3: Average accuracy of VitMTSC and other models on UEA datasets. A dash indicates that the corresponding dataset was not evaluated in the respective paper. The **VitMTSC(val)** column shows model overfit, since it was also used for hyper-parameter tuning.

| DATASET | CMFMTS | TST | ROCKET | VitMTSC(val) | VitMTSC(test) |
|---------|--------|-----|--------|--------------|---------------|
| PenDigits | 95.9 | - | 97.8 | **99.4** | 97.2 |
| SpokenArabicDigits | 97.5 | **99.8** | 99.8 | 99.0 | 96.9 |
| CharacterTrajectories | 97.1 | - | **99.3** | 97.1 | 98.5 |
| FaceDetection | 57.9 | 68.9 | 64.0 | **72.7** | 67.0 |
| InsectWingbeat | 67.7 | **68.7** | 53.6 | 60.4 | 60.3 |

As we can see, model overfits validation set since it was used for both hyper-parameter tuning and guide training but generalized poorly on 3 out of 5 unseen test datasets - especially on smaller datasets. We hope that future MTSC papers will follow our approach and won't use provided test dataset for

---

[4]https://www.sktime.org/en/latest/index.html

either training or hyper-parameter tuning. This will help benchmark generalization capabilities of MTSC models correctly. Since Zerveas et al. (2021) has already explored impact of pre-training on UEA datasets, we didn't. Rather, in future, we would like to explore pre-training methods that works well on ViT e.g., BEiT (Bao et al., 2022).

## 4.4 OFFLINE EXPERIMENTS ON COMMERCIAL DATASETS

We compared our model against several DNN time-series classifiers that learns, like VitMTSC, end-to-end directly from raw MTS data without feature engineering. For comparative analysis, we implemented and evaluated six published MTSC DL models — FCN (Wang et al., 2016), Inception Time (Fawaz et al., 2019), ResCNN (Zou et al., 2019), ResNet (Fawaz et al., 2018), CropTransformer (Rußwurm et al., 2019) and TST (Zerveas et al., 2021) — and the VitMTSC model on our smallest dataset (refer Table 2). We divided this dataset into 595K (336K +ve and 259K -ve) training, 198K (112K +ve and 86K -ve) validation and 198K (112K +ve and 86K -ve) test sets. Our time-series sample consisted of 10-300 datapoints depending on user activities. Since our data is unbalanced, we use class weights to account for the imbalance. Also, as mentioned in Section 3.1.3, we do not use absolute or learnable positional encodings and instead include the temporal information in our multi-variate data - timestamp encoding. Table 4 shows the PRAUC for the different models. Based on these results, we trained only our VitMTSC model for our larger datasets with 6M datapoints.

Table 4: PRAUC of VitMTSC vs different DL models on data from one of our smaller stores. The model predicts whether a customer will respond/not respond to a marketing campaign.

| MODEL | RESPOND | NOT RESPOND |
|---|---|---|
| FCN | 0.57 | 0.47 |
| Inception Time | 0.56 | 0.42 |
| ResCNN | 0.55 | 0.41 |
| ResNet | 0.54 | 0.44 |
| CropTransformer | 0.55 | 0.45 |
| TST | 0.56 | 0.45 |
| **VitMTSC** | **0.96** | **0.95** |

## 4.5 ONLINE EXPERIMENT ON COMMERCIAL DATASETS

In an online A/B test between VitMTSC and an existing heuristic-based baseline method for finding potential customers who will respond to a marketing campaign, the conversion rate (the percentage of customers who responded to the campaign out of those who were targeted) went up by an average of 352% (p-value $< 0.001$).

### 4.5.1 ABLATION STUDY ON COMMERCIAL DATASETS

We performed four ablations to evaluate the importance of placement of layer-normalization, padding masks, class weights and positional encoding (refer Table 5). Though we performed the ablations on nine datasets, for simplicity, we present results from only two datasets (refer Table 2) with the understanding that other datasets followed similar trends. Section 3.1.5 and Section 4.6 provides detailed information on Layer-Normalization and its impact. As also noticed by Nguyen & Salazar (2019), even with learning rate warmup, the VitMTSC PostNorm version failed to converge on our datasets. Adding explicit positional encoding deteriorates performance. This is because positional encoding adds an arbitrary order to multiple activities that occurred at the same time. Instead using the categorical time features (month, day, hour, etc) improves the performance of the model. The importance of class-weight depends on the amount of data. Only in the Large dataset where the data size is an order of magnitude more than other datasets, class-weighting was not required. All other datasets (eight stores - though Table 5 report results on only one of the eight stores and the largest store) showed improved performance with class weights. Similarly, masking allowed the classifier to ignore irrelevant (padded) data-points and improved performance. Table 5 gives the average PRAUC of the VitMTSC with different configurations on these real-world commercial datasets.

Table 5: VitMTSC ablation study: average PRAUC on two commercial datasets. We test out (a) Pre and Post layer-normalization, (b) using learnable positional encoding vs timestamp positional encoding (including the temporal information as features in our multi-variate data), (c) using class weights and (d) padding masks. Each ablation uses the best values from the other three combinations.

| DATASET | LAYER-NORM | | POSITION-ENCODING | | CLASS-WEIGHT | | MASK | |
| | PRE | POST | LEARNABLE | TIMESTAMP | YES | NO | YES | NO |
| --- | --- | --- | --- | --- | --- | --- | --- | --- |
| Small | **0.961** | 0.574 | 0.943 | **0.961** | **0.961** | 0.955 | **0.961** | 0.957 |
| Large | **0.975** | 0.528 | 0.949 | 0.971 | 0.971 | **0.975** | 0.971 | 0.969 |

## 4.6 COMPARISON OF OTHER MTSC TRANSFORMER MODELS TO VITMTSC

CropTransformer (Rußwurm et al., 2019) and TST (Zerveas et al., 2021) are other two Transformer models for MTSC. The CropTransformer official implementation uses Rectified Linear Unit (RELU) (Nair & Hinton, 2010) activation function, *POSTNORM* layer-normalization (Ba et al., 2016), and hand-crafted absolute positional encoding like the original Transformer (Vaswani et al., 2017). TST uses *POSTNORM* like the original transformer but replaces layer-normalization with batch-normalization (Ioffe & Szegedy, 2015), learnable positional encoding and GELU activation function. VitMTSC on the other hand, uses *PRENORM* layer-normalization, GELU activation function and learnable positional and timestamp encoding (refer to 3.1.3). Also, similar to BERT's [CLS] and ViT's [class] token, VitMTSC uses only [class] token at output layer for MLP and discards rest of transformer encoder output. In contrast, both CropTransformer and TST don't use [CLS] token and feed full output of transformer encoder, in the end, to the MLP layer for classification. This [class] token provides an aggregate representation of MTS sequence to the final MLP Head for classification. Table 4, and Table 5 show impact of these choices in model performance on commercial datasets ( 4.1.2). Both official implementations of TST[5] and CropTransformer[6] are in PyTorch and PyTorch offical transformer encoder implementation added support for PreNorm(default=False) in mid-2021[7]. For this reason, we decided to provide the Algorithm for VitMTSC ( 3.1.6).

The main reason why VitMTSC significantly outperform CropTransformer (Rußwurm et al., 2019) and TST (Zerveas et al., 2021) on real-world datasets is: VitMTSC, similar to ViT, uses *PRENORM* layer-normalization(refer to 3.1.5). Nguyen & Salazar (2019); Xiong et al. (2020) provide detailed insights of impact of *PRENORM* vs *POSTNORM* in Transformer. In our ablation study on using *PRENORM* and *POSTNORM* (refer Table 5), we found that *PRENORM* is the key to learning on larger commercial datasets (refer Table 2). As mentioned in Section 4.5.1 and also noticed by Nguyen & Salazar (2019), even with learning rate warmup, the VitMTSC PostNorm version failed to converge on our datasets.

## 5 CONCLUSION AND FUTURE WORK

We introduce Vision Transformer for Multivariate Time-Series Classification (VitMTSC), a novel deep learning architecture for multivariate time-series classification (MTSC) based on ViT (Dosovitskiy et al., 2020). We demonstrate, on open-source datasets, VitMTSC's performance matches the current SOTA methods, the model scales to large datasets seen in commercial retail sites and provides far superior performance than other neural net based MTSC models. Since, the VitMTSC model can learn directly from raw multivariate time-series data, we can avoid the manual feature engineering, which is a fundamental part of non-DNN MTSC algorithms.

Finally, we emphasize the significance of *PRENORM* Layer normalization in transformer networks. Attempts have been made to comprehend ViT's success (Raghu et al., 2021; Trockman & Kolter, 2022) and enhance its performance further (Chen et al., 2021). As part of further research, we will investigate effect of *PRENORM/POSTNORM* Layer normalization on ViT, since we observed significant impact in our commercial MTS datasets.

---

[5]`https://github.com/gzerveas/mvts_transformer`
[6]`https://github.com/dl4sits/BreizhCrops`
[7]`https://github.com/pytorch/pytorch/issues/55270`

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

# A APPENDIX

## A.1 EXPERIMENT DETAILS

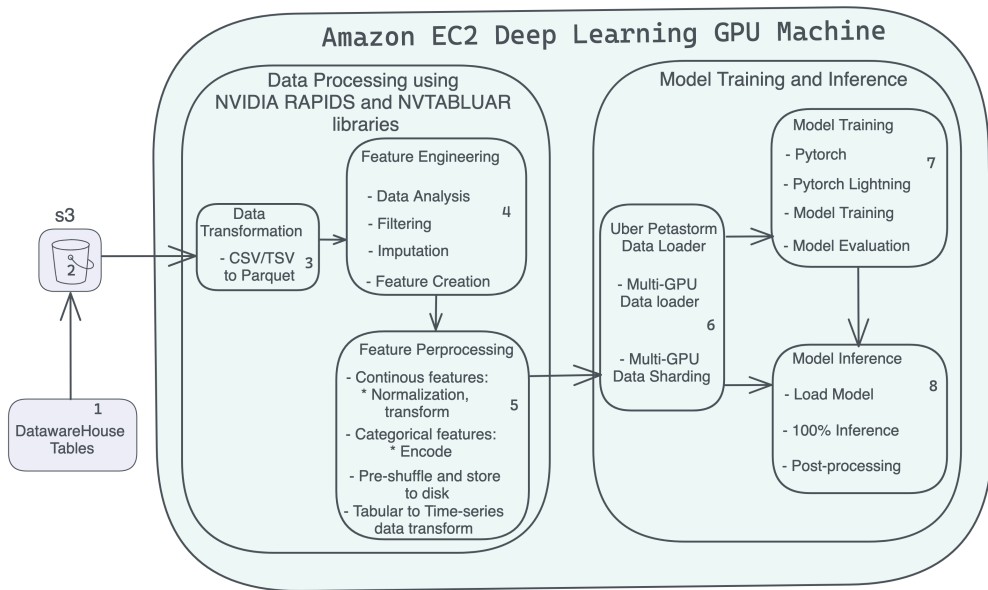

Figure 3: VitMTSC deep learning training and inference pipeline

### A.1.1 TOOLS

To efficiently process terabytes (TB) of commercial MTSC data, we use Graphical Processing Units (GPUs) for end-to-end to accelerated data processing as well as model training/inference (Raschka et al., 2020). We use a combination of open-source libraries: a) NVIDIA RAPIDS and NVTabular (NVIDIA, 2018; 2021) GPU accelerated libraries for data-processing and feature engineering; b) Uber Petastorm (Petastorm, 2021) based data-loaders for multi-gpu training/inference; c) Pytorch-Lightning (Lightning, 2021) for model training/inference; d) nbdev (fast.ai, 2020) to enable code sharing across Jupyter notebooks; and e) Conda (Conda, 2020) and Ploomber (Ploomber, 2021) for workflow automation. With help of these tools, we are able to write end-to-end ML pipeline in Jupyter and test it locally or deploy it to Cloud. To reproduce our experiment results on open-source UEA datasets, we provide the fully self-contained code in supplemental material[8].

### A.1.2 ENVIRONMENT

We run all experiments locally on Amazon Elastic Compute Cloud (EC2) p3dn.24xlarge machine or deployed full workflow remotely on Amazon Web Service (AWS) Batch service on g4dn.4xlarge machines. All machines have AWS Deep Learning Amazon Machine Images (DLAMI) and Ubuntu operating system. The p3dn.24xlarge machine has 96 CPUs, 768GB RAM and 256GB of total GPU memory divided across eight GPU cores. Similarly, g4dn.4xlarge machine has 16 CPUs, 64GB RAM and 16GB of total GPU memory on a single GPU core.

### A.1.3 TECHNIQUES

The open-source UEA datasets (refer Table 1) used a single NVIDIA Tesla V100 GPU core whereas commercial datasets (refer Table 5) used all eight NVIDIA Tesla V100 GPU cores for both training and inference. We use the Extract step of Extract-Transform-Load (ETL) to pull data from Dataware House (DW) to Amazon Simple Storage Service (Amazon S3) and use GPU workflow (Figure 3) for the remaining steps of the ML modeling.

---

[8]To facilitate further research, we release code at `https://github.com/mtsc-research/vitmtsc`.

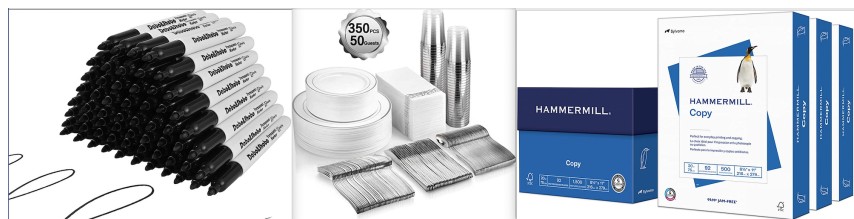

Figure 4: Top 3 products - all business-related bulk purchases - identified by VitMTSC model indicate that the user will be responsive to a marketing campaign targeted towards small businesses.

As shown in Figure 3, we keep data in tabular format until last stage of **Step 5** for data-processing and feature engineering. And then we convert tabular data to time-series format, such that, all data related to a single time-series are in a single row in parquet file. This helps Uber Petastorm (Petastorm, 2021) based data-loaders to load all data for a time-series in one read during multi-gpu training/inference.

## A.2 INSPECTING VITMTSC

Transformer self-attention layers assign an attention value between each pair of tokens. In NLP, a token is typically a word or a word-part. For images, each token represents an image patch, while in the VitMTSC model, each token represents a single MTSC datapoint in the time series. Each token is linked to the [class] token via the self-attention mechanism. The strength of the attention link can intuitively indicate each token's contribution to the classification. However, the attention values only represent a single facet of the Transformer network. Additionally, non-positive activation functions and skip connections impede accurate interpretation. Chefer et al. (2020) proposed attributing relevance to features in Transformer models by introducing a relevancy propagation rule that is applicable for both positive and negative transformations, normalizing relevance for non-parametric layers such as "add" (e.g., skip-connection) and matrix multiplication, integrating the attention and relevancy scores, and combining the integrated results for multiple attention blocks.

We adapted the ViT implementation of the Chefer et al. (2020) to VitMTSC and demonstrated interpretable model outcomes. We focus on the k time-series datapoints that contributed the most to the prediction. This method identifies the top k time-series data points in a sample that contribute to a specific classification. Figure 4, for instance, depicts the top three product purchases that were significant for classifying whether a user would be receptive to a small business-oriented marketing campaign. Note that each of the three purchases consists of bulk business-related products.

## A.3 UEA DATASETS

The UEA MTSC archive is a repository of benchmark datasets for time-series classification (Bagnall et al., 2018). It includes 30 multivariate datasets, four of which have variable lengths. We test our model on the following five datasets (Table 1) available in the UEA repository, which include three datasets of variable length, a condition frequently observed in real-world data.

### A.3.1 PENDIGITS

A UCI Archive dataset (Dua & Graff, 2017; Alimoglu, 1996). This is a handwritten digit classification task. 44 writers were asked to draw the digits [0...9], where instances are made up of the x and y coordinates of the pen traced across a digital screen. The data was originally recording these coordinates at a 500x500 pixel resolution, however was normalised and sampled to 100x100. Then, on expert knowledge from the dataset's authors, the data was spatially resampled such that instead of each consecutive attribute having a constant time step (100ms) but variable spatial step, they instead had constant spatial step and variable time step. From experimentation by the authors, the data was resampled to 8 spatial points, such that each instance is 2 dimenions of 8 points, with a single class label [0...9] being the digit drawn.

### A.3.2 SPOKENARABICDIGITS

This data set is taken from the UCI repository (Dua & Graff, 2017; Hammami & Sellam, 2009). It is derived from sound. Dataset from 8800 (10 digits x 10 repetitions x 88 speakers) time series of 13 Frequency Cepstral Coefficients (MFCCs) had taken from 44 males and 44 females Arabic native

speakers between the ages 18 and 40 to represent ten spoken Arabic digit. Each line on the data base represents 13 MFCCs coefficients in the increasing order separated by spaces. This corresponds to one analysis frame.

### A.3.3 CHARACTERTRAJECTORIES

From the UCI archive (Dua & Graff, 2017). The characters here were used for a PhD study on primitive extraction using HMM based models. The data consists of 2858 character samples. The data was captured using a WACOM tablet. 3 Dimensions were kept - x, y, and pen tip force. The data has been numerically differentiated and Gaussian smoothed, with a sigma value of 2. Data was captured at 200Hz. The data was normalised. Only characters with a single 'PEN-DOWN' segment were considered. Character segmentation was performed using a pen tip force cut-off point. The characters have also been shifted so that their velocity profiles best match the mean of the set. Each instance is a 3-dimensional pen tip velocity trajectory. The original data has different length cases. The class label is one of 20 characters 'a', 'b', 'c', 'd', 'e', 'g', 'h', 'l', 'm', 'n', 'o', 'p', 'q', 'r', 's', 'u', 'v', 'w', 'y', 'z'. To conform with the repository, we have truncated all series to the length of the shortest, which is 182, which will no doubt make classification harder.

### A.3.4 FACEDETECTION

This data comes from a Kaggle competition in 2014 (Kaggle, 2017). The problem is to determine whether the subject has been shown the picture of a face or a scrambled image based on MEG, independent of the subject. This data set is just the training data from the competition, split by patient into 10 train subjects (subject01 to subject10) and 6 test subjects (subject11 to 16). This data cannot sensibly be randomly resampled. A leave one patient out approach is the most suitable. There are about 580-590 trials per patient, giving 5890 train trials and 3524 test subjects. Each trial consists of 1.5 seconds of MEG recording (starting 0.5sec before the stimulus starts) and the related class label, Face (class 1) or Scramble (class 0). The data were down-sampled to 250Hz and high-pass filtered at 1Hz, giving 62 observations per channel. 306 timeseries were recorded, one for each of the 306 channels, for each trial.

### A.3.5 INSECTWINGBEAT

The InsectWingbeat data was generated by the UCR computational entomology group (Chen & Adena Why, 2017) and used in the paper "Flying Insect Classification with Inexpensive Sensors. Journal of Insect Behaviour 2014" (Chen et al., 2014). The original data is a reconstruction of the sound of insects passing through a sensor. The data in the archive is the power spectrum of the sound. The 10 classes are male and female mosquitos (Ae. aegypti, Cx. tarsalis, Cx. quinquefasciants, Cx. stigmatosoma), two types of flies (Musca domestica and Drosophila simulans) and other insects. A spectorgram of each 1 second sound segment was created with a window length of 0.061 seconds and an overlap of 70%. Each instance in this multivariate dataset is arranged such that each dimension is a frequency band from the spectrogram. Each of the 10 classes in this dataset consist of 5,000 instances.

### A.4 PYTORCH/PYTORCH LIGHTNING CODE FOR VITMTSC

### A.4.1 IMPORTS

```python
import torch
import pytorch_lightning as pl
from torch import nn
from torch.nn import functional as F
from torchmetrics import functional as FM
from petastorm import make_batch_reader
from petastorm.pytorch import DataLoader
from einops import rearrange, repeat
```

### A.4.2 RESIDUAL CONNECTION

```python
class Residual(nn.Module):
    def __init__(self, fn):
        super().__init__()

        self.fn = fn
```

```python
    def forward(self, x, **kwargs):
        return self.fn(x, **kwargs) + x
```

### A.4.3 PreNorm: Layer-Normalization

```python
class PreNorm(nn.Module):
    def __init__(self, dim, fn):
        super().__init__()

        self.norm = nn.LayerNorm(dim)
        self.fn = fn

    def forward(self, x, **kwargs):
        return self.fn(self.norm(x), **kwargs)
```

### A.4.4 Feed Forward Network sub-Layer

```python
class FeedForward(nn.Module):
    def __init__(self, dim, hidden_dim, dropout=0.0):
        super().__init__()

        self.net = nn.Sequential(
            nn.Linear(dim, hidden_dim),
            nn.GELU(),
            nn.Dropout(dropout),
            nn.Linear(hidden_dim, dim),
            nn.Dropout(dropout),
        )

    def forward(self, x):
        return self.net(x)
```

### A.4.5 Multi-Head Attention sub-Layer

```python
class Attention(nn.Module):
    def __init__(self, dim, heads=10, dim_head=32, dropout=0.0):
        super().__init__()

        inner_dim = dim_head * heads
        self.heads = heads
        self.scale = dim_head ** -0.5

        self.to_qkv = nn.Linear(dim, inner_dim * 3, bias=False)
        self.to_out = nn.Sequential(nn.Linear(inner_dim, dim),
        ↪   nn.Dropout(dropout))

    def forward(self, x, mask=None, register_hook=False):
        b, n, _, h = *x.shape, self.heads
        qkv = self.to_qkv(x).chunk(3, dim=-1)
        q, k, v = map(lambda t: rearrange(t, "b n (h d) -> b h n
        ↪   d", h=h), qkv)

        dots = torch.einsum("bhid,bhjd->bhij", q, k) * self.scale
        mask_value = -torch.finfo(dots.dtype).max

        if mask is not None:
            mask = F.pad(mask.flatten(1), (1, 0), value=True)
            mask = mask.unsqueeze(1).unsqueeze(2)

            assert mask.shape[-1] == dots.shape[-1], "mask has
            ↪   incorrect dimensions"
            dots.masked_fill_(mask == 0.0, mask_value)
```

```python
            del mask

        attn = dots.softmax(dim=-1)
        out = torch.einsum("bhij,bhjd->bhid", attn, v)

        out = rearrange(out, "b h n d -> b n (h d)")
        out = self.to_out(out)
        return out
```

### A.4.6 TRANSFORMER ENCODER

```python
class Transformer(nn.Module):
    def __init__(self, dim, depth, heads, dim_head, mlp_dim,
    ↪  dropout):
        super().__init__()

        self.layers = nn.ModuleList([])
        for _ in range(depth):
            self.layers.append(nn.ModuleList([
                Residual(PreNorm(dim, Attention(dim, heads =
                ↪  heads, dim_head = dim_head, dropout =
                ↪  dropout))),
                Residual(PreNorm(dim, FeedForward(dim, mlp_dim,
                ↪  dropout = dropout)))
            ]))

    def forward(self, x, mask = None, register_hook = False):
        for attn, ff in self.layers:
            x = attn(x, mask = mask, register_hook =
            ↪  register_hook)
            x = ff(x)
        return x
```

### A.4.7 VITMTSC MODEL

```python
class VitMTSCModel(pl.LightningModule):
    def __init__(self, config, c_in = NUMBER_OF_FEATURES, c_out =
    ↪  NUM_TARGET,
                seq_len = SEQUENCE_LENGTH,class_weight =
                ↪  torch.FloatTensor(CLASS_WEIGHT)):
        super(VitMTSCModel, self).__init__()

        self.d_model = config["d_model"]
        self.depth = config["depth"]
        self.heads = config["heads"]
        self.mlp_dim = config["mlp_dim"]
        self.dim_head = config["dim_head"]
        self.dropout_p = config["dropout"]
        self.emb_dropout_p = config["emb_dropout"]
        self.lr = config["lr"]
        self.weight_decay = config["weight_decay"]
        self.patience = config["patience"]

        self.pos_embedding = nn.Parameter(torch.randn(1, seq_len +
        ↪  1, self.d_model))
        self.patch_to_embedding = nn.Linear(c_in, self.d_model)
        self.cls_token = nn.Parameter(torch.randn(1, 1,
        ↪  self.d_model))
        self.dropout = nn.Dropout(self.emb_dropout_p)
```

```python
        self.transformer = Transformer(self.d_model, self.depth,
        ↪    self.heads, self.dim_head, self.mlp_dim,
        ↪    self.dropout_p)
        self.to_cls_token = nn.Identity()
        self.mlp_head = nn.Sequential(
            nn.LayerNorm(self.d_model),
            nn.Linear(self.d_model, self.mlp_dim),
            nn.GELU(),
            nn.Dropout(self.dropout_p),
            nn.Linear(self.mlp_dim, c_out)
        )

        self.c_out = c_out
        self.register_buffer('class_weight', class_weight)

    def forward(self, x, mask = None, register_hook = False):
        x = self.patch_to_embedding(x) # bs x seq_len x nvars ->
        ↪    bs x seq_len x d_model
        b, n, _ = x.shape # bs, seq_len

        cls_tokens = repeat(self.cls_token, '() n d -> b n d', b =
        ↪    b) # bs x 1 x d_model
        x = torch.cat((cls_tokens, x), dim=1) # bs x (seq_len + 1)
        ↪    x d_model
        x += self.pos_embedding[:, :(n + 1)] # +=  1 x (seq_len +
        ↪    1) x d_model -> # bs x (seq_len + 1) x d_model
        x = self.dropout(x) # bs x (seq_len + 1) x d_model

        x = self.transformer(x, mask = mask, register_hook =
        ↪    register_hook) # bs x (seq_len + 1) x d_model

        x = self.to_cls_token(x[:, 0]) # bs x d_model
        return self.mlp_head(x) # bs x num_classes

    def configure_optimizers(self):
        optimizer = torch.optim.AdamW(self.parameters(),
        ↪    lr=self.lr, weight_decay=self.weight_decay)
        scheduler =
        ↪    torch.optim.lr_scheduler.ReduceLROnPlateau(optimizer,
        ↪    patience=self.patience)
        return {"optimizer": optimizer, "lr_scheduler": scheduler,
        ↪    "monitor": "train_loss"}

    def training_step(self, batch, batch_idx):
        x, y, _, mask = batch
        y_hat = self(x, mask)
        y = y.long()
        train_loss = F.cross_entropy(y_hat, y, weight =
        ↪    self.class_weight)
        train_auc =  FM.accuracy(F.softmax(y_hat, dim=1), y,
        ↪    num_classes = self.c_out)
        self.log('train_loss', train_loss, on_step=False,
        ↪    on_epoch=True, prog_bar=True, logger=True)
        self.log('train_auc', train_auc, on_step=False,
        ↪    on_epoch=True, prog_bar=True, logger=True)
        return train_loss

    def validation_step(self, batch, batch_idx):
        x, y, _, mask = batch
```

```
        y_hat = self(x, mask)
        y = y.long()
        val_loss = F.cross_entropy(y_hat, y, weight =
        ↪   self.class_weight)
        val_auc =  FM.accuracy(F.softmax(y_hat, dim=1), y,
        ↪   num_classes = self.c_out)
        self.log('val_loss', val_loss, on_step=False,
        ↪   on_epoch=True, prog_bar=True, logger=True,
        ↪   sync_dist=True)
        self.log('val_auc', val_auc, on_step=False, on_epoch=True,
        ↪   prog_bar=True, logger=True)
        return val_loss

    def test_step(self, batch, batch_idx):
        x, y, _, mask = batch
        y_hat = self(x, mask)
        y = y.long()
        test_loss = F.cross_entropy(y_hat, y, weight =
        ↪   self.class_weight)
        test_auc =  FM.accuracy(F.softmax(y_hat, dim=1), y,
        ↪   num_classes = self.c_out)
        self.log('test_loss', test_loss, on_step=False,
        ↪   on_epoch=True, prog_bar=True, logger=True,
        ↪   sync_dist=True)
        self.log('test_auc', test_auc, on_step=False,
        ↪   on_epoch=True, prog_bar=True, logger=True)
        return test_loss
```

### A.4.8   VITMTSC: PETASTORM DATAMODULE

```
def petastorm_collate_fn(rows):
    data_df = pd.DataFrame(rows)
    case_id_df = data_df.iloc[:,
    ↪   NUMBER_OF_FEATURES*SEQUENCE_LENGTH+1
        :NUMBER_OF_FEATURES*SEQUENCE_LENGTH+2]
    case_id_tensor =
    ↪   torch.tensor(case_id_df.values.astype(np.float64))

    target_df = data_df.iloc[:,
    ↪   NUMBER_OF_FEATURES*SEQUENCE_LENGTH+0
        :NUMBER_OF_FEATURES*SEQUENCE_LENGTH+1]
    target_tensor =
    ↪   torch.tensor(target_df.values.astype(np.float32))

    data_tensor_df = data_df.iloc[:, 0*SEQUENCE_LENGTH
        :NUMBER_OF_FEATURES*SEQUENCE_LENGTH]
    data_tensor =
    ↪   torch.tensor(data_tensor_df.values.astype(np.float32))
    data_tensor = rearrange(data_tensor, 't (b h)-> t h b', h =
    ↪   SEQUENCE_LENGTH)

    mask_df = data_df.iloc[:, 0*SEQUENCE_LENGTH
        :1*SEQUENCE_LENGTH]
    mask_tensor = torch.tensor(mask_df.values.astype(np.float32))

    return data_tensor, target_tensor.squeeze(),
    ↪   case_id_tensor.squeeze(), mask_tensor.squeeze()

class VitMTSCPetastormDataModule(pl.LightningDataModule):
    def __init__(
```

```python
        self,
        config,
        train_files,
        valid_files,
        test_files,
        num_workers=NUM_WORKERS,
        transform_spec=None,
        shard_count=NUM_GPUS,
        num_epochs=MAX_EPOCHS,
    ):
        super().__init__()
        self.train_files = train_files
        self.valid_files = valid_files
        self.test_files = test_files
        self.batch_size = config["batch_size"]
        self.num_workers = num_workers
        self.transform_spec = transform_spec
        self.shard_count = shard_count
        self.num_epochs = num_epochs

    def train_dataloader(self):

        self.train_ds = make_batch_reader(
            self.train_files,
            workers_count=self.num_workers,
            transform_spec=self.transform_spec,
            cur_shard=int(os.environ["LOCAL_RANK"]),
            shard_count=self.shard_count,
            num_epochs=self.num_epochs,
        )
        return DataLoader(self.train_ds,
        ↪  batch_size=self.batch_size,
        ↪  collate_fn=petastorm_collate_fn)

    def val_dataloader(self):
        print(
            "val_dataloader: local rank :",
            int(os.environ["LOCAL_RANK"]),
            "shard count: ",
            self.shard_count,
        )
        self.val_ds = make_batch_reader(
            self.valid_files,
            workers_count=self.num_workers,
            transform_spec=self.transform_spec,
            cur_shard=int(os.environ["LOCAL_RANK"]),
            shard_count=self.shard_count,
            num_epochs=self.num_epochs,
        )
        return DataLoader(self.val_ds, batch_size=self.batch_size,
        ↪  collate_fn=petastorm_collate_fn)

    def test_dataloader(self):
        print(
            "test_dataloader: local rank :",
            int(os.environ["LOCAL_RANK"]),
            "shard count: ",
            self.shard_count,
        )
```

```
self.test_ds = make_batch_reader(
    self.test_files,
    workers_count=self.num_workers,
    transform_spec=self.transform_spec,
    cur_shard=int(os.environ["LOCAL_RANK"]),
    shard_count=self.shard_count,
    num_epochs=self.num_epochs,
)
return DataLoader(self.test_ds,
↪  batch_size=self.batch_size,
↪  collate_fn=petastorm_collate_fn)
```

