# OpenReview forum: "VISION TRANSFORMER FOR MULTIVARIATE TIME- SERIES CLASSIFICATION (VITMTSC)"
_ICLR.cc/2023/Conference — Submitted to ICLR 2023_

### Official Review · Reviewer_ivET · 2022-10-24

**Confidence:** 4
**Correctness:** 3
**Technical Novelty And Significance:** 2
**Empirical Novelty And Significance:** 2
**Recommendation:** 5

**Clarity, Quality, Novelty And Reproducibility:**

The idea is quite simple and obtain promising results on real-world commercial datasets. However, it's unclear why VitMTSC can outperform state-of-the-art methods in a large margin in real-world commercial datasets while slightly left behind on public benchmarks. Without further insights, it's hardly novel in term of both technical and empirical perspectives. In addition, code link seems to be empty although authors claim that it's publicly available.

**Strength And Weaknesses:**

Strength
* The idea is quite simple and cast time series as an "Image" and apply ViT to time series classification tasks.
* Results on real-word commercial datasets are significantly better than baselines.

Weaknesses
* The idea is hardly novel and it's more like a pure application of Vision Transformer without any deep insight.
* It's unclear why VitMTSC is significantly better than state-of-the-art baselines on commercial datasets. Authors don't discuss these insights in their papers and it's unclear when practitioners should utilize vision transformer for time series classification.
* The accuracy is pretty high on PenDigits, SpokenArabicDigits and CharacterTrajectories. Is it due to data imbalance? Could authors provide other metric, e.g. F1?
* This paper mainly explores binary classifications and multiple-class classification, however, is not fully considered.

**Summary Of The Paper:**

This paper applies VisionTransformer for Multivariate Time-Series Classification (VitMTSC) model that learns latent features from raw time-series data for classification tasks and is applicable to large-scale time-series data with millions of data samples of variable lengths. Their results show that it's can obtain near state-of-the-art results on public benchmark and obtain significant improvements on real-world commercial datasets.

**Summary Of The Review:**

This paper applies vision transformer in Multivariate Time-Series Classification problems. Results show that it can outperform state-of-the-art on commercial datasets while slightly left behind on public benchmark datasets. It's unclear why this performance gap exist, making this paper less valuable.

---

> ### Author Response · Authors · 2022-11-19
> **Revised paper to address feedback**
>
> We thank the reviewer for the thoughtful feedback, questions and comments, which give us the opportunity to improve the paper. We have updated and uploaded revised paper. Here is the brief summary of our response:
>
> * Open-source UEA datasets have multiple classes. The  CLASSES column of Table 1 shows number of different classes in each dataset.
> * We will upload code to github.com once our names are revealed as part of review. However, we have already attached the code in the *Supplementary Material along with initial paper submission.*
> * In section 4.1, we discuss in detail difference between open-source UEA datasets and private commercial datasets to the extent allowed by our organization. In re-written Section 4.6, we have explained why ViT works for MTSC as compared to other Transformer based MTSC models.
> * VitMTSC novelty:
>     * The MTS data is already in flattened 1-D patch (Section 3.1.1). Different modalities will need different pre-processing before it’s ready to be fed to ViT e.g., ViViT: A Video Vision Transformer (https://arxiv.org/pdf/2103.15691.pdf) needs different pre-processing than ViT. Similarly, MTS data needs pre-processing mentioned in section 3.1.1-3.1.3.
>     * In re-written Section 4.6, we have explained why ViT works for MTSC as compared to other Transformer based MTSC models.
>     * We also re-wrote Section 4.3, to better explain current practice in MTSC research publication and put VitMTSC reported model performance in context.

---

### Official Review · Reviewer_fdnF · 2022-10-25

**Confidence:** 3
**Correctness:** 2
**Technical Novelty And Significance:** 1
**Empirical Novelty And Significance:** 2
**Recommendation:** 3

**Clarity, Quality, Novelty And Reproducibility:**

The clarity and reproducibility of this paper is in OK level, most of the notations can match up, and the figures are informative. Details of this model are well discussed, including various aspects like feature encoding, padding, position embedding. The Algorithm in page 6 looks a bit confusing because there are some notations unintroduced. Experiment section includes both UEA dataset and real world application, where there's not enough in-depth analysis on the gap of analysis between them. The most concerning point comes from the design of this architecture, where the concept "patch" of Vision Transformer was nowhere to be seen. It seems the whole input processing and embedding process is just a feature engineering for Transformer input. The use of `PRENORM` seems interesting but it's already proposed in GPT-2 (Radford et al., 2019) and ViT (Dosovitskiy et al., 2020). Therefore, the paper seems to lack novelty and hard to find impact other than the performance on the real-world application dataset, which I'm not sure if it's gonna be open sourced. Therefore, the quality and novelty of this paper is low.


**Strength And Weaknesses:**

This paper provides an interesting idea of using a Vision Transformer for a Multivariate Time-Series Classification problem. The strengths of the paper are:
1. The figures are well-made and improve the clarity of the model description.
2. Details on tokenization, feature encoding, padding and truncation are discussed in the paper, improving the paper's reproducibility.
3. This paper did some interesting ablation study on layer normalization that change its position from after `MSA/FNN block + Residual connection` which is also called `POSTNORM` to before `MSA/FFN block` which is also called `PRENORM`. Experiments show it can improve applications' performance greatly on real-world dataset.
4. The experiments section includes some real world dataset, including offline and online tests, which is interesting to see.

On the other hand, the weaknesses of the paper are:

1. Even though the author(s) claimed this paper is based on Vision Transformer, there's no concept of "patch" applied in this feature extraction stage. The input was naturally given in different time stamps and features were extracted in a one-by-one fashion. After tokenization and feature encoding, there's only a linear projection applied onto each of the feature vectors. I tend to consider this process as some kind of feature engineering for the input of a Transformer. Can you please further explain how this is related to vision transformer's patch concept and whether the novelty comes from this process?
2. Covariates (time features in this paper) like month, day, hour have proven to be very useful in time series, while using them as position encoding could help, have you tried combining it into the original feature vector before linear projection and combine with learnable position encoding?
3. In the UEA dataset, the performance of the proposed model looks a bit worse than state-of-the-art models, but performance a lot better then these models on real world dataset. There seems to be not enough case study and analysis to further understand where the difference comes from and how we can mitigate the gap. I suggest the author(s) to provide more insights on the cause of this difference.
4. The bolded numbers in tables are a bit confusing, for example in Table 3, VitMTSC's validation set results were bolded but other results are from the test set. In Table 5, some of the results in the same ablation study groups are bolded but some are not, e.g., the `padding mask` group where the dataset is `Large`. Can you please add some explanation on the bolded number meaning?
5. The author(s) claimed to have `experimented with different aggregation strategies` but there were no conclusions or results mentioned. Can you please show the conclusion or performance with different aggregation strategies' impact?


**Summary Of The Paper:**

This paper proposed the `VitMTSC` model, a vision Transformer model that learns latent features from raw time-series data for classification tasks, and could be applied on large-scale time series data with variable lengths. The model can reach comparable performance on UEA datasets against previous state-of-the-art methods and outperform them by a large margin on real-world dataset.

The contributions of this paper are from proposing a VitTransformer based approach to model long-range contextual relationship in MTS data, and demonstrate its superior performance on real-world application data.


**Summary Of The Review:**

To summarize, this paper introduced a Vision Transformer based model called `VitMTSC` to improve multivariate time series classification. However, the novelty in this paper seems to be limited and the results have a huge gap between open-source UEA dataset and real-world application dataset. Even though the performance on real-world dataset looks great and the model can handle large scale dataset, the engineering part seems out of the scope of this paper and there's a lot more to explain on the experiment results. Therefore, I recommend this paper to be rejected.

---

> ### Author Response · Authors · 2022-11-19
> **Revised paper to address feedback**
>
> We thank the reviewer for the thoughtful feedback, questions and comments, which give us the opportunity to improve the paper. We have updated and uploaded revised paper. Here is the brief summary of our response:
>
> * The MTS data is already in flattened 1-D patch (Section 3.1.1). Different modalities will need different pre-processing before it’s ready to be fed to ViT e.g., ViViT: A Video Vision Transformer (https://arxiv.org/pdf/2103.15691.pdf) needs different pre-processing than ViT. Similarly, MTS data needs pre-processing mentioned in section 3.1.1-3.1.3.
>   * Patch is not novelty for VitMTSC.  Patches Are All You Need? (https://openreview.net/forum?id=TVHS5Y4dNvM) investigated role of patch embedding in success of ViT and “authors cannot directly prove this concept "patch is the most critical component" yet.  Instead in MTSC domain, we found PreNorm is playing most significant role in success of our model when compared to other Transformer based MTSC models (TST, CorpTransformer). We have re-written Section 4.6 to better explain why our model works.
>
>  * As discussed in 3.1.3 Positional Encoding sub-section, some datasets can have a set of (unordered) data points with the same timestamp. For example, for time-ordered set of products purchased by a customer, a single customer cart (same timestamp) may include multiple products. In such a case, absolute or learnable positional encodings will impose an arbitrary order on these datapoints (products).
>     *  That’s said, combining time-features into the original feature vector before linear projection and combine with learnable BERT like sentence embedding, instead of learnable position encoding , will be more suitable for this use-case. We will explore this approach in future.
>
> * We have re-written Section 4.3 to better explain model performance as compared to other SOTA models.
> * Fixed Table 3 and re-wrote Section 4.3. In Table 5, each ablation uses the best values from the other three combinations.
> * In our commercial applications, in addition to MTS data, we have time-invariant sequence level side-information (e.g., user attributes). To add side-information to the ViTMTSC transformer model, we used different aggregation strategies proposed in paper ~\citep{Moreira2021Transformers4RecBT}. Since, open-source UEA datasets doesn’t provide side-information, we have deleted this line from paper to avoid confusion.
> * The  https://arxiv.org/pdf/2207.09238.pdf paper provided pseudocode for the original Transformer and its encoder/decoder-only variations. The essentially Transformer complete pseudocode is about 50 lines. One may argue that most DL models are minor variations of a few core architectures, such as the Transformer.
>
>        * In re-written Section 4.6, we have explained why previous approaches to apply Transformer doesn’t work on large commercial datasets. Also, in section 3.1.1-3.1.3.  we have explained how we prepare MTS data for Transformer Encoder and explained how Target Encoding to encode categorical features help keep model complexity low.
> * We have shared the full code of end-to-end pipeline, in supplementary section.

---

### Official Review · Reviewer_YY3h · 2022-10-28

**Confidence:** 4
**Correctness:** 2
**Technical Novelty And Significance:** 2
**Empirical Novelty And Significance:** 2
**Recommendation:** 3

**Clarity, Quality, Novelty And Reproducibility:**

The paper is clear, of high quality, and reproducible. The novelty is limited considering that it's applying an existing model.

**Strength And Weaknesses:**

S

- extensive experiments with multiple datasets and setups
- straightforward application of ViT to timeseries

W

- lack of discussion wrt better results in baselines (TST, Rocket)
- limited novelty compared to existing models (ViT)

**Summary Of The Paper:**

The paper proposes the application of ViT models for the task of timeseries classification.

**Summary Of The Review:**

It's unclear whether this model improves against other sota models in the literature based on Table 3. It seems that baselines perform equally well or even better here. The proposed model seems to perform better only on the internal dataset (private?) which makes the comparison with public benchmarks really difficult. I would urge the authors to expand more on the public benchmark experimentation and discuss (and expand) Table 3.

---

> ### Author Response · Authors · 2022-11-19
> **Revised paper to address feedback**
>
> We thank the reviewer for the thoughtful feedback and comments, which give us the opportunity to improve the paper. We have updated and uploaded revised paper. Here is the brief summary of our response:
>
> * We have re-written section 4.3 and section 4.6 to better explain our results compared to other MTSC models.
> * Section 3.1 and re-written Section 4.6 explains VitMTSC novelty. Also, Section 4.6  now explains why VitMTSC significantly outperforms other Transformer based MTSC models.
> * As suggested, we have re-written section 4.3 that better explains Table 3.

---

### Decision · Program_Chairs · 2023-01-20

**Decision:**

Reject

**Justification For Why Not Higher Score:**

The paper lacks novelty in method, and is a direct demonstration of ViT to several time-series data sets. At least a theory on why the ViT works superior to baseline methods or vice-versa would have enhanced novelty.

**Justification For Why Not Lower Score:**

The authors can enhance the novelty and resubmit the manuscript to another venue. Hope the review comments are useful for revision.

**Metareview: Summary, Strengths And Weaknesses:**

The paper is an interesting application of the vision transformer for time-series data set, but has conducted extensive experiments to demonstrate the effectiveness on time-series data sets

The Novelty is limited and the superiority of the ViT to baseline methods, or the superiority of the baseline methods on some data sets are not adequately discussed.

**Summary Of Ac-Reviewer Meeting:**

NIL